https://doi.org/10.1038/s42003-023-05317-9　　OPEN
# Calpeptin is a potent cathepsin inhibitor and drug candidate for SARS-CoV-2 infections

Patrick Y. A. Reinke [1,22], Edmarcia Elisa de Souza[2,22], Sebastian Günther [1,22], Sven Falke [1], Julia Lieske [1], Wiebke Ewert [1], Jure Loboda [3,4], Alexander Herrmann [5], Aida Rahmani Mashhour [1], Katarina Karničar [3,6], Aleksandra Usenik [3,6], Nataša Lindič [3], Andreja Sekirnik[3], Viviane Fongaro Botosso [7], Gláucia Maria Machado Santelli[8], Josana Kapronezai [7], Marcelo Valdemir de Araújo [7,9], Taiana Tainá Silva-Pereira[9,10], Antônio Francisco de Souza Filho [9], Mariana Silva Tavares[9], Lizdany Flórez-Álvarez[2], Danielle Bruna Leal de Oliveira [9], Edison Luiz Durigon[9], Paula Roberta Giaretta [11], Marcos Bryan Heinemann [10], Maurice Hauser[12], Brandon Seychell[13], Hendrik Böhler[13], Wioletta Rut[14], Marcin Drag [14], Tobias Beck [13,15], Russell Cox [12], Henry N. Chapman [1,15,16], Christian Betzel [15,17], Wolfgang Brehm[1], Winfried Hinrichs [18], Gregor Ebert [5,19], Sharissa L. Latham[20,21], Ana Marcia de Sá Guimarães [9], Dusan Turk [3,6✉], Carsten Wrenger [2✉] & Alke Meents [1✉]

Several drug screening campaigns identified Calpeptin as a drug candidate against SARS-CoV-2. Initially reported to target the viral main protease (M$^{pro}$), its moderate activity in M$^{pro}$ inhibition assays hints at a second target. Indeed, we show that Calpeptin is an extremely potent cysteine cathepsin inhibitor, a finding additionally supported by X-ray crystallography. Cell infection assays proved Calpeptin's efficacy against SARS-CoV-2. Treatment of SARS-CoV-2-infected Golden Syrian hamsters with sulfonated Calpeptin at a dose of 1 mg/kg body weight reduces the viral load in the trachea. Despite a higher risk of side effects, an intrinsic advantage in targeting host proteins is their mutational stability in contrast to highly mutable viral targets. Here we show that the inhibition of cathepsins, a protein family of the host organism, by calpeptin is a promising approach for the treatment of SARS-CoV-2 and potentially other viral infections.

[1] Center for Free-Electron Laser Science CFEL, Deutsches Elektronen-Synchrotron DESY, Notkestr. 85, 22607 Hamburg, Germany. [2] Department of Parasitology, Institute of Biomedical Sciences at the University of São Paulo, São Paulo, Brazil. [3] Department of Biochemistry and Molecular and Structural Biology, Jozef Stefan Institute, Jamova 39, 1000 Ljubljana, Slovenia. [4] Jožef Stefan International Postgraduate School, Jamova 39, Ljubljana, Slovenia. [5] Institute of Virology, Helmholtz Munich, Munich, Germany. [6] Centre of Excellence for Integrated Approaches in Chemistry and Biology of Proteins, Jamova 39, 1000 Ljubljana, Slovenia. [7] Virology Laboratory, Center of Development and Innovation, Butantan Institute, São Paulo, Brazil. [8] Department of Cell and Developmental Biology, Institute of Biomedical Sciences, University of São Paulo, São Paulo, Brazil. [9] Department of Microbiology, Institute of Biomedical Sciences, University of São Paulo, São Paulo, Brazil. [10] Department of Preventive Veterinary Medicine and Animal Health, College of Veterinary Medicine, University of São Paulo, São Paulo, Brazil. [11] Gastrointestinal Laboratory, Department of Small Animal Clinical Sciences, College of Veterinary Medicine and Biomedical Sciences, Texas A&M University, 4474 TAMU School Station, TX, USA. [12] Institute for Organic Chemistry and BMWZ, Leibniz University of Hannover, Schneiderberg 38, 30167 Hannover, Germany. [13] Department of Chemistry, Institute of Physical Chemistry, Universität Hamburg, Grindelallee 117, 20146 Hamburg, Germany. [14] Department of Chemical Biology and Bioimaging, Faculty of Chemistry, Wrocław University of Science and Technology, Wybrzeże Wyspiańskiego 27, 50-370 Wrocław, Poland. [15] Hamburg Centre for Ultrafast Imaging, Universität Hamburg, Luruper Chaussee 149, 22761 Hamburg, Germany. [16] Department of Physics, Universität Hamburg, Luruper Chaussee 149, 22761 Hamburg, Germany. [17] Department of Chemistry, Institute of Biochemistry and Molecular Biology and Laboratory for Structural Biology of Infection and Inflammation, c/o DESY, Universität Hamburg, 22607 Hamburg, Germany. [18] Universität Greifswald, Institute of Biochemistry, Felix-Hausdorff-Str. 4, 17489 Greifswald, Germany. [19] Institute of Virology, Technical University of Munich, Munich, Germany. [20] The Kinghorn Cancer Centre, Garvan Institute of Medical Research, Sydney, NSW, Australia. [21] St Vincent's Hospital Clinical School, UNSW, Sydney, NSW, Australia. [22] These authors contributed equally: Patrick Y. A. Reinke, Edmarcia Elisa de Souza, Sebastian Günther. ✉email: dusan.turk@ijs.si; cwrenger@icb.usp.br; alke.meents@desy.de

Three years after its outbreak, the severe acute respiratory syndrome coronavirus 2 (SARS-CoV-2) pandemic has an ongoing impact on human health. Due to continuously emerging escape mutants and the occurrence of long coronavirus disease (COVID) cases, there remains an urgent need for antiviral drugs in addition to continuously adapted vaccines, and two approved drugs targeting the virus main protease (MPro)[1-3]. In several screening efforts, calpain inhibitors such as Calpeptin have been identified as promising drug candidates against COVID-19[4-6]. Although first identified as compounds acting directly against MPro, calpain inhibitors have more recently been reported to interfere with virus entry into host cells[5,7-10]. This latter effect has been attributed to the efficient inhibition of cysteine cathepsins by calpain inhibitors[11]. In contrast to viral proteins, proteins of the host organism, which are essential for infecting cells, have so far played a minor role in antiviral drug development. Yet in comparison to the highly mutable viral drug and vaccine targets, targeting host proteins appears advantageous due to their negligible variability in the human population[2]. The inhibition of essential host proteins generally bears a risk of side effects. However, an acute life-threatening viral infection is different from a long-term chronic disease such as osteoporosis. Nevertheless, advantages and disadvantages of such an approach must be carefully considered.

In a first common infection step of the host organism, the viral Spike-protein (S-protein) of SARS-CoV-2 and related viruses binds to the host cell receptor angiotensin converting-enzyme 2 (ACE2)[12-14], followed by two different cell entry pathways (Fig. 1). In the case of the cell surface pathway, the S-protein can be proteolytically activated by the type II transmembrane serine protease (TMPRSS2)[14,15]. In the case of the endosomal pathway, the SARS-CoV-2 viral particles enter the cell via clathrin-mediated endocytosis followed by proteolytic cleavage of S-protein by Cathepsin L (CatL)[16,17], which enhances virion release from endolysosomes and elicits productive viral infections[13,18]. In addition to CatL, other cysteine cathepsins have been reported to be essential for processing the viral S-protein upon cell entry[19-22] (Fig. 1).

In a previously performed large-scale X-ray crystallography screen, we identified Calpeptin as a binder of the active site of the SARS-CoV-2 main protease[4]. In subsequent VERO E6 cell-based SARS-CoV-2 infection assays, Calpeptin showed a high suppression of virus replication at concentrations below 100 nM. This is amongst the lowest reported values in such assays and, in combination with its moderate inhibition of MPro in biochemical assays, hints at an additional target. Such a dual-targeting approach has been reported for several calpain inhibitors[4,9,10,23-25]. Among these, Calpeptin in particular has demonstrated promising properties as a SARS-CoV-2 drug candidate[5,26].

Here, we have further investigated a potential dual-targeting effect of Calpeptin and, indeed, show the pico-molar inhibition of cathepsins by Calpeptin, its sulfonated prodrug analog S-Calpeptin, and the chemically similar prodrug GC-376 (Table 1)[24]. Binding of the activated compounds to MPro, CatL, CatK, and CatV was characterized by X-ray crystallography and their antiviral activity against SARS-CoV-2 was confirmed in cellular infection assays. Finally, the most promising compound S-Calpeptin was tested as a COVID-19 treatment in a hamster model.

## Results

To test the potential dual-targeting of MPro and cysteine cathepsins by Calpeptin, we compared the inhibition of various cathepsins and MPro by Calpeptin and the chemically similar prodrug GC-376 (Table 1). Calpeptin and GC-376 differ in the replacement of norleucine in Calpeptin with a 2-oxopyrrolidine side chain in GC-376. In addition, the aldehyde warhead is sulfonated in GC-376, which is expected to increase the stability in the organism by protecting the aldehyde warhead from metabolism as demonstrated for Norovirus 3CLpro inhibitors[27] and, as additional advantage, to increase its solubility. At physiological conditions the sulfonated compound slowly releases the hydrogensulfite, thereby unmasking the aldehyde warhead (Fig. S1). Due to the expected improved pharmacokinetics by masking the

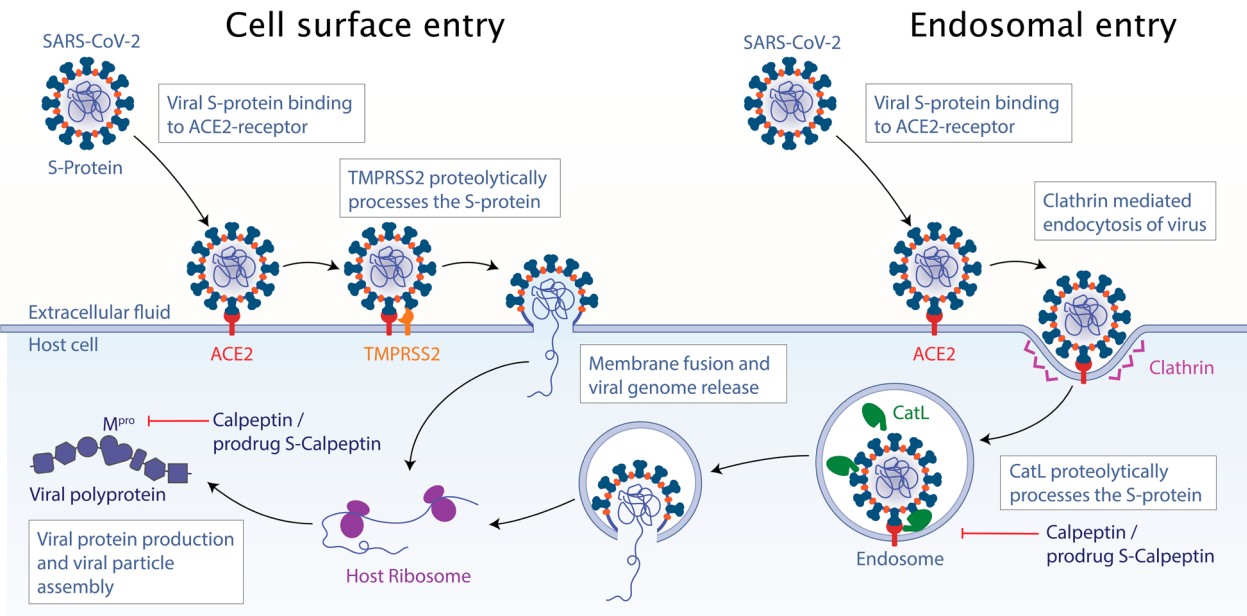

**Fig. 1 Cell entry pathways of SARS-CoV-2.** After initial virus binding with its S protein to the ACE2 receptor, cell entry proceeds via two pathways. In case of cell surface entry, the S-protein is cleaved by TMPRSS2, inducing membrane fusion and viral genome release. In case of the endosomal entry, the viral particle is internalized by clathrin-mediated endocytosis and subsequently the S-protein is activated by CatL in the endosome, followed by membrane fusion and genome release.

**Table 1 Chemical structure of compounds and their in vitro inhibition properties.**

| Inhibitor | SARS-CoV-2 M$^{pro}$ $K_i$ [nM] | CatB $K_i$ [nM] | CatK $K_i$ [pM] | CatL $K_i$ [pM] | CatV $K_i$ [pM] | VERO E6 EC$_{50}$ [nM] |
|---|---|---|---|---|---|---|
| Calpeptin | 4500 (3900–5600) | 41 ± 7/27–54 | 61 ± 12/37–85 | 131 ± 21/90–172 | 361 ± 47/268–454 | 183 ± 65 |
| S-Calpeptin | 4700 (1100–8000) | 70 ± 15/39–100 | 50 ± 12/28–73 | 148 ± 19/111–185 | 169 ± 18/133–204 | 120 ± 54 |
| GC-376 | <100 [40][a] | 163 ± 50/63–262 | 91 ± 11/70–112 | 259 ± 27/204–314 | 242 ± 23/196–288 | 1071 ± 278 |

$K_i$ values for SARS-CoV-2 M$^{pro}$ inhibition are mean with 95% confidence interval. Values for cathepsin inhibition are mean with standard deviation and 95% confidence interval for $K_i$ determination and mean with standard error for EC50 determination in VERO E6 cells.
[a]$K_i$ as reported previously[66].

warhead, we included the sulfonated form of Calpeptin into our experiments[27].

**Enzyme inhibition assays.** Calpeptin, S-Calpeptin, and GC-376 were tested in protease inhibition assays of human cathepsins and SARS-CoV-2 M$^{pro}$ (Table 1). The lowest $K_i$ values are observed for the inhibition of CatK and CatL by Calpeptin with 61 pM and 131 pM, respectively. S-Calpeptin shows similar inhibition properties (50 and 148 pM, respectively). For GC-376, slightly weaker inhibition was found (91 pM and 259 pM). For all three compounds, $K_i$ values for CatV are similar to CatL, whereas CatB values are generally in the nanomolar range. In contrast to CatL and CatK inhibition, which is in the picomolar range, the inhibition of M$^{pro}$ is in the micromolar range for Calpeptin and nanomolar range for GC-376. This difference in cathepsin and M$^{pro}$ inhibition implies that several cysteine cathepsins, all endopeptidases, are inhibited much more efficiently within the cellular context than M$^{pro}$ at the same inhibitor concentration.

**X-ray structure analysis.** Binding of Calpeptin and the activated forms of the prodrugs S-Calpeptin and GC-376 to both CatL and M$^{pro}$ were independently confirmed by X-ray crystallography. Moreover, Calpeptin binding was also confirmed in a CatK and CatV structure (Table 2). All three compounds bind in a highly similar manner to the catalytic site of each protease (Fig. S2a–h). On the structural basis, M$^{pro}$ has a more constrained binding pocket leading to a high sequence specificity[28], in contrast the active site of CatL is less constrained (Fig. S2a, b), resulting in a more promiscuous substrate specificity reflecting its role in protein degradation in the endosomal pathway[29] (Fig. 2).

The active site of CatL is formed by two loops of the L-domain (Gln$^{21}$–Cys$^{25}$ and Gly$^{61}$–Leu$^{69}$) interconnected by a disulfide bond and two loops from the R-domain (Ile$^{136}$–His$^{163}$, Tyr$^{182}$–Ala$^{212}$) (Fig. 2a–c)[30]. The catalytic dyad of the active site are Cys$^{25}$ in the L-domain and His$^{163}$ in the R-domain. Both Calpeptin and the GC-376 aldehyde bind covalently via the aldehyde group to Cys$^{25}$, forming a thiohemiacetal, exhibiting the same binding mode (Fig. 2b, d, e). From the S-Calpeptin derived binding of the desulfonated compound Calpeptin is observed in the same binding pose (Figs. 2d and S12). The three amino acid analogs of these peptidomimetic inhibitors occupy the substrate sites S1 to S3 (Fig. 2d). The main interactions of Calpeptin and the GC-376 aldehyde are mediated by the peptide backbone forming hydrogen bonds to the CatL main chain residues Gly$^{68}$ and Asp$^{162}$ in a substrate-like manner (Fig. 2b). In addition, the thiohemiacetal oxygen forms a hydrogen bond to Gln$^{19}$. The only chemical difference between the GC-376 aldehyde and Calpeptin is the replacement of the norleucine side chain with 2-oxopyrrolidine (Fig. 2d, e). In the Calpeptin structure the norleucine side chain forms hydrophobic contacts to Gly$^{23}$ and Asn$^{66}$. The analog 2-oxopyrrolidine group of GC-376 shows also hydrophobic interactions with these residues. The central leucine side chains of both compounds are covered in the hydrophobic S2-pocket formed by Leu$^{69}$, Met$^{70}$ and Ala$^{135}$. The benzyloxycarbonyl-groups (Cbz-group) interact via hydrophobic contacts to Glu$^{63}$ and Gly$^{67}$. In both structures the terminal Cbz-group is located at the S3 position. It should be noticed that small molecules like PEG fragments of the crystallization solution were found in the S1'-pocket. No differences in the binding mode of Calpeptin to the highly similar CatK and CatV are observed (Fig. S12). Binding of the GC-376 aldehyde and Calpeptin to M$^{pro}$ is similar and specific variations are described in Supplementary Notes 1 and 2.

**In vitro cell culture experiments with S-Calpeptin.** Initial cell culture SARS-CoV-2 infection assays using VERO E6 cells with

**Table 2 Structures of enzyme/inhibitor-complexes.**

| PDB ID Protein/Compound | 7Z3T CatL | 7QKC CatL/S-Calp | 7Z58 CatL/Calpeptin | 7QKB CatL/GC-376 | 7Z3U Mpro/S-Calp | 7QKA Mpro/GC-376 | 7QGW CatV/Calp | 8C3D CatK/S-Calp |
|---|---|---|---|---|---|---|---|---|
| **Data collection** | | | | | | | | |
| Space group | P1 | P1 | P1 | P1 | P 21 21 21 | C 1 2 1 | P43 21 2 | C 1 2 1 |
| Cell dimensions a, b, c (Å) | 56.94, 62.14, 67.47 | 57.05, 62.46, 67.54 | 57.18, 62.45, 67.82 | 57.29, 62.45, 67.82 | 67.692, 99.598, 103.261 | 113.675, 53.356, 44.994 | 94.24, 94.24, 126.96 | 94.63, 41.62, 65.66 |
| α, β, γ (°) | 105.166, 94.226, 115.576 | 105.32, 93.91, 115.39 | 105.578, 93.134, 115.562 | 105.306, 93.523, 115.929 | 90, 90, 90 | 90, 102.511, 90 | 90, 90, 90 | 90, 121.01, 90.0 |
| Resolution (Å)[a] | 49.94 - 1.6 (1.657 - 1.6) | 44.27 - 1.69 (1.752 - 1.691) | 49.64 - 1.35 (1.398 - 1.35) | 44.25 - 1.8 (1.864 - 1.8) | 49.22 - 1.72 (1.782 - 1.72) | 48.09 - 1.8 (1.864 - 1.8) | 47.12 - 1.3 (1.35 -1.303) | 34.94 - 2.0 (2.12 - 2.00) |
| $R_{merge}$[a] | 0.1661 (2.183) | 0.1705 (1.244) | 0.2549 (1.335) | 0.1346 (0.9518) | 0.101 (2.172) | 0.04322 (0.53) | 0.0548 (1.023) | 0.100 (0.232) |
| $R_{meas}$[a] | 0.1793 (2.345) | 0.1809 (1.378) | 0.2568 (1.364) | 0.1476 (1.115) | 0.1086 (2.328) | 0.04674 (0.5756) | 0.05597 (1.057) | 0.110 (0.270) |
| $R_{pim}$[a] | 0.06553 (0.839) | 0.05976 (0.579) | 0.03067 (0.2767) | 0.05975 (0.5693) | 0.03923 (0.8283) | 0.01761 (0.2213) | 0.0113 (0.2616) | 0.0449 (0.1304) |
| Mean $I/sigma\,(I)$[a] | 9.07 (2.33) | 9.33 (1.57) | 16.83 (2.11) | 9.62 (1.99) | 12.54 (0.95) | 23.40 (3.05) | 32.94 (2.56) | 23.32 (11.35) |
| CC 1/2[a] | 0.99 (0.55) | 0.996 (0.567) | 0.999 (0.835) | 0.994 (0.73) | 0.999 (0.524) | 1 (0.9) | 1 (0.837) | 0.997 (0.961) |
| Completeness (%)[a] | 97.22 (96.66) | 98.06 (83.93) | 99.98 (99.90) | 90.98 (81.98) | 99.70 (98.31) | 98.89 (98.22) | 99.91 (99.16) | 98.9 (96.2) |
| Redundancy[a] | 7.2 (7.6) | 8.7 (5.2) | 61.1 (23.1) | 5.0 (3.3) | 7.5 (7.7) | 6.9 (6.6) | 23.7 (16.0) | 5.89 (4.45) |
| **Refinement** | | | | | | | | |
| Resolution (Å)[a] | 49.94 - 1.6 (1.657 - 1.6) | 44.27 - 1.691 (1.752 - 1.691) | 49.64 - 1.35 (1.398 - 1.35) | 44.25 - 1.8 (1.864 - 1.8) | 49.22 - 1.72 (1.782 - 1.72) | 48.09 - 1.8 (1.864 - 1.8) | 47.12 - 1.303 (1.35 -1.303) | 34.94 - 2.0 (2.03 - 2.00) |
| No. reflections[a] | 100909 (10010) | 87087 (7453) | 175655 (17545) | 67329 (6065) | 74542 (7273) | 24243 (2368) | 139006 (13653) | 14932 (682) |
| $R_{work}$[a]/$R_{free}$ | 0.1547 (0.2424)/ 0.1760 (0.2612) | 0.1465 (0.2700)/ 0.1790 (0.3048) | 0.1300 (0.2262)/0.1574 (0.2515) | 0.1694 (0.3041)/ 0.2030 (0.3546) | 0.1865 (0.3802)/ 0.2157 (0.3986) | 0.1629 (0.2237)/ 0.2072 (0.3017) | 0.1718 (0.2454)/ 0.1956(0.2775)[b] | 0.1959 (0.2268) / 0.2405 (0.1887)[b] |
| No. non-hydrogen atoms | 7801 | 7645 | 8359 | 7279 | 5442 | 2711 | | 1913 |
| Protein | 7057 | 6822 | 7161 | 6785 | 4911 | 2417 | 3479 | 1649 |
| Ligand/ion | 140 | 153 | 230 | 222 | 94 | 70 | 196 | 26 |
| Water | 586 | 670 | 968 | 272 | 437 | 224 | 2040 | 238 |
| B-factors | 29.23 | 27.28 | 20.09 | 52.57 | 41.32 | 36.95 | 25.89 | 11.37 |
| Protein | 28.5 | 26.1 | 18.17 | 52.2 | 40.85 | 36.95 | 19.42 | 9.42 |
| Ligand/ion | 48.56 | 50.48 | 38.12 | 65.86 | 53.36 | 40.68 | 56.23 | 19.49 |
| Water | 33.44 | 34 | 29.99 | 51.17 | 44.02 | 39.66 | 53.05 | 24.96 |
| R.m.s. deviations | | | | | | | | |
| Bond lengths (Å) | 0.005 | 0.009 | 0.011 | 0.006 | 0.006 | 0.013 | 0.017 | 0.018 |
| Bond angles (°) | 0.88 | 1.26 | 1.08 | 0.65 | 0.92 | 1.08 | 1.83 | 2.05 |
| Ramachandran favored | 97.78 | 97.56 | 97.58 | 96.84 | 98.1 | 98.02 | 97.26 | 96.7 |
| allowed | 2.22 | 2.44 | 2.42 | 3.16 | 1.9 | 1.98 | 2.74 | 3.3 |
| outliers | 0 | 0 | 0 | 0 | 0 | 0 | 0 | 0 |

X-ray data collection and refinement statistics for enzyme/inhibitor complexes.
[a]Values in parentheses are for the highest-resolution shell.
[b]$R_{kick}$ instead of $R_{free}$.

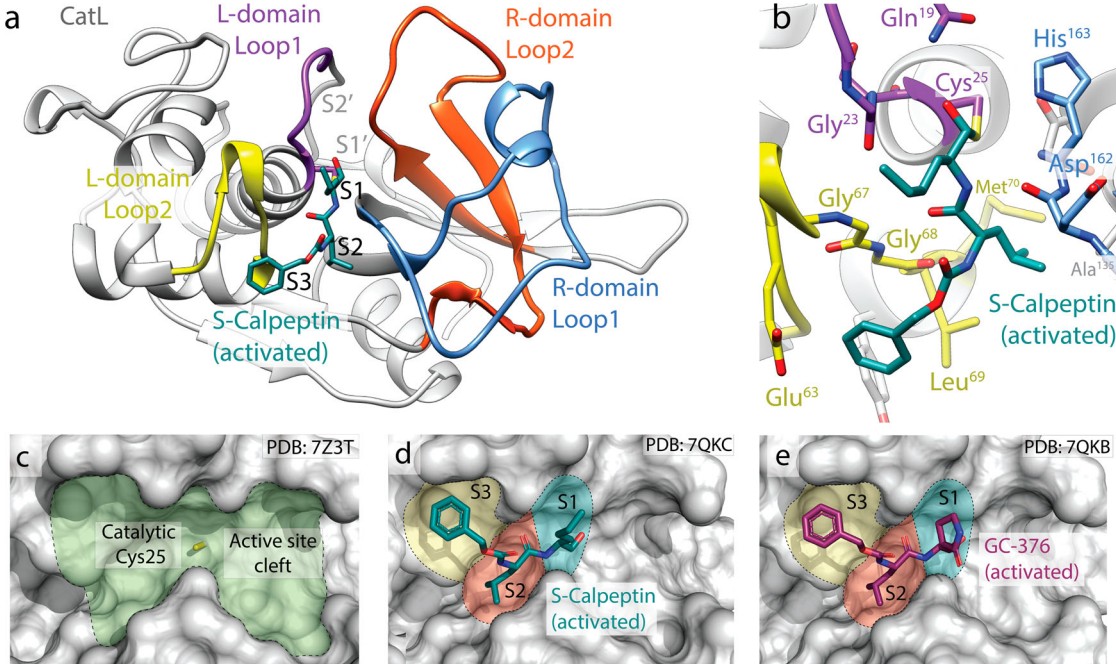

**Fig. 2 Inhibitor binding to the active site of CatL. a** Domain structure of CatL with Calpeptin derived from prodrug S-Calpeptin bound to the active site. **b** Close-up view of the active site of CatL with bound S-Calpeptin. Residues involved in the catalytic mechanism are highlighted in stick representation. **c** Surface representation of the empty active site of CatL with the catalytic cysteine highlighted in yellow. Calpeptin derived from S-Calpeptin (**d**) and the GC-376 aldehyde (**e**) bound to the CatL active site with highlighted subsites S1–S3.

all three compounds indicated that the inhibition potential of Calpeptin and S-Calpeptin is much higher compared to GC-376 (Table 1 and Fig. S3). Therefore, we focused our further investigations on Calpeptin and its prodrug S-Calpeptin. Both were tested for inhibition of SARS-CoV-2 infections at the cellular level. We have used two cell lines for SARS-CoV-2 infection VERO E6 and the closely related VERO-CCL81 cells[31]. While the former does not express TMPRSS2, the latter expresses TMPRSS2 at a very low level[17] (Fig. S4 and Supplementary Note 3). Besides these two established cell lines for SARS-CoV-2 infection we have used human non-small cell lung carcinoma LC-HK2 to more closely mimic the cells that SARS-CoV-2 would first encounter when introduced into the lung and enable both the surface and endosomal entry pathway of the virus. For this cell line we were able to detect strong TMPRSS2 expression in comparison with VERO-CCL81 (Fig. S4). Calpeptin and S-Calpeptin show no cytotoxicity up to low micromolar concentrations in both cell-lines (Fig. 3) and no substantial impact on vesicle morphology in LC-HK2 cells (Fig. S5a–c and Supplementary Note 4).

Inhibition of SARS-CoV-2[32] infection as detected by mRNA levels was found to be comparably effective in VERO-CCL81 (Calpeptin $EC_{50}$ of 0.99 μM, S-Calpeptin $EC_{50}$ of 0.7 μM) and in LC-HK2 (Calpeptin $EC_{50}$ of 1.7 μM, S-Calpeptin $EC_{50}$ of 0.63 μM) (Fig. 3). Assays testing the inhibition of the cytopathic effect (CPE) revealed an almost tenfold increase of effectiveness for LC-HK2 (Calpeptin $EC_{50}$ of 0.06 μM, S-Calpeptin $EC_{50}$ of 0.07 μM), compared to VERO-CCL81 (Calpeptin $EC_{50}$ of 0.4 μM, S-Calpeptin $EC_{50}$ of 0.7 μM) (Fig. 4).

**Animal experiments**. Due to the promising properties of Calpeptin in our experiments, in a next step we tested its effectiveness in suppressing SARS-CoV-2 infections in vivo in Golden Syrian hamsters. Hamsters have been identified as an animal model that is susceptible to SARS-CoV-2 infections and that resembles the lung pathology of human patients[33]. For animal testing, we used the Calpeptin prodrug S-Calpeptin, for which we expect improved pharmacokinetics similar to GC-376[27,34,35].

Hamsters were intranasally infected with $1 \times 10^5$ TCID50 (50% Tissue Culture Infectious Dose) of SARS-CoV-2, strain B.1.1.28, on day 0 (Fig. S6a). On day 1, treatment with S-Calpeptin was started at a dose of 1 mg/kg body weight, which was determined in a pre-experiment (Supplementary Note 5). Body weight and clinical signs were followed daily (Fig. S6a, b) with animals euthanized on days 3, 5, and 7 post-infection (p.i.) to analyze viral load and histopathological lesions in respiratory tissues. Overall, treated infected and untreated infected animals showed similar weight changes (Fig. S6c) and respiratory signs such as snout rubbing and sneezing between days 2 and 5. Histopathology analysis showed no statistically significant differences in lesion scores between SARS-CoV-2 infected animals that were treated with S-Calpeptin and to those that were not (Figs. S7a-i and S8a–f and Supplementary Note 6). Lung vascular lesions are a hallmark of COVID-19 in hamsters[36]. Although S-Calpeptin-treated animals showed lower scores for vascular damage compared to the untreated group on days 5 and 7 p.i., these results were not statistically significant (Fig. S8e). Analysis of the viral load revealed lower SARS-CoV-2 titers in the treated group compared to the untreated group on day 5 p.i. for both lungs and trachea, with a statistically significant difference on TCID50 viral load in trachea (Fig. 5).

## Discussion

Our experiments demonstrate that the prodrug S-Calpeptin and its active form Calpeptin are highly potent inhibitors of CatL and other cathepsins with activity in the picomolar range. In contrast, Calpeptin and the related GC-376 in its active aldehyde form show at least three orders of magnitude lower activity on SARS-CoV-2 $M^{pro}$, suggesting that cysteine cathepsins are their primary targets. Cathepsins in general, and CatL in particular, are known to play a central role in the endosomal entry pathway of SARS-CoV-2 into host cells. Calpeptin is a broad-band cysteine protease inhibitor covalently binding with its aldehyde group to the active

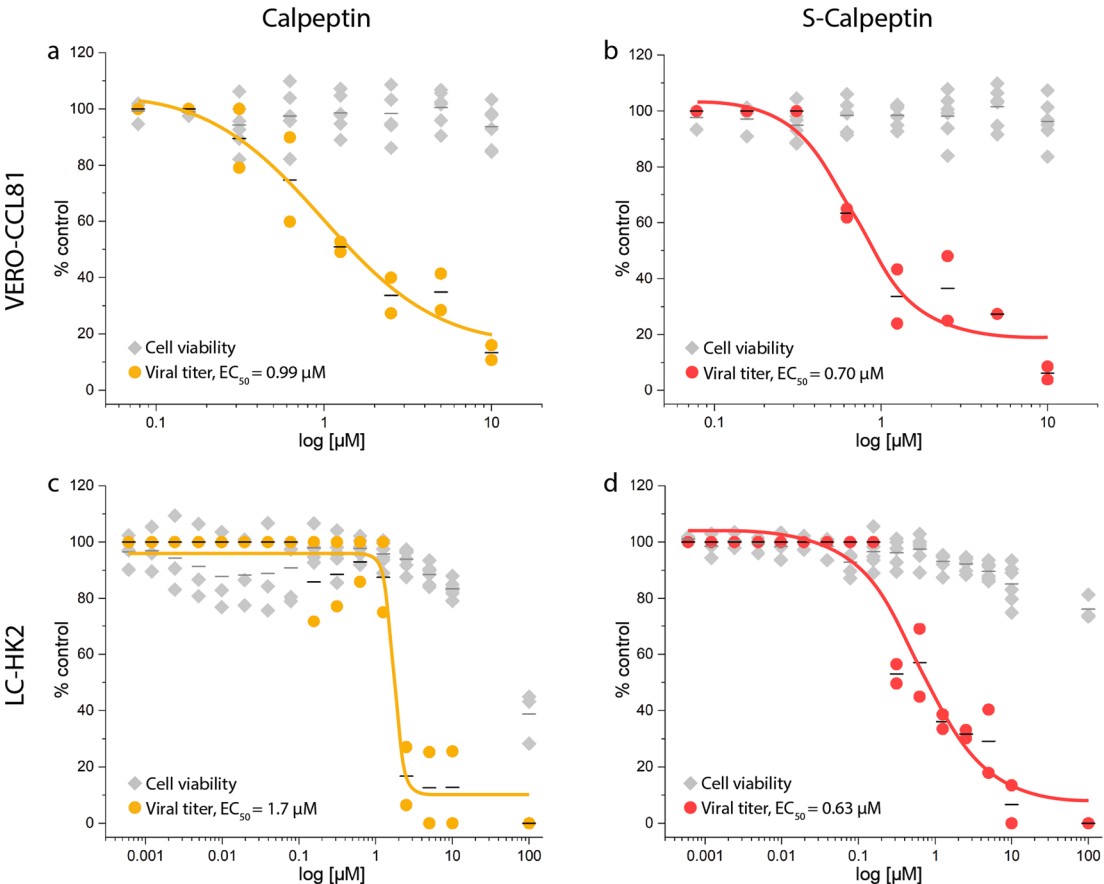

**Fig. 3 Calpeptin and S-Calpeptin inhibit SARS-CoV-2 infection.** The viral titer (●) and the cell viability (◆) of VERO-CCL81 (**a**, **b**) and human LC-HK2 (**c**, **d**) cells were determined by RT-qPCR and CellTiter-Glo luminescence method, respectively. EC$_{50}$ values for viral titers are shown. Individual data points and the mean (——) of three replicates of two biological experiments are displayed as line. EC$_{50}$ was calculated by fitting the data using GraphPad Prism version 8.0.

site cysteine of CatL. Even in TMPRSS2 expressing cell lines, i.e. in the presence of the CatL-independent entry pathway, Calpeptin was effective in suppressing SARS-CoV-2 infections. These findings were confirmed in SARS-CoV-2-infected hamsters, treatment with the prodrug S-Calpeptin led to a significant reduction of the viral load in trachea on day 5. Taken together, these findings clearly indicate the high potential of Calpeptin and its prodrug S-Calpeptin as treatment for SARS-CoV-2 infections.

Calpeptin was first described in 1988 and was shown to strongly penetrate cell membranes[37]. While it was developed to explore the role of calpain and its effects in the cell, several studies have described it to interact with various disease pathways such as neuropathy, Parkinson's disease, and inflammation[38–40]. In general, the anti-inflammatory effect of Calpeptin could be an additional effect that helps against COVID-19 and especially against long COVID-19, which has been recently discussed in the literature[26]. Therefore, the use of calpain inhibitors against COVID-19 is increasingly the focus of current research, as they are often found in high-throughput screens[4,5,41].

Due to its broad-band inhibition of proteases, Calpeptin is expected to cause several side effects during daily treatment. The dosage chosen for our hamster studies was a daily dose of 1 mg/kg s.c. over seven days with no apparent side effects, while higher doses (2 mg/kg or 3 mg/kg) resulted in blood leukopenia at the end of the seven-days protocol. The modest reductions in viral loads observed in vivo could be attributed to this limitation in dosage to avoid side effects. It is possible that the distribution and levels of S-Calpeptin in the tissue were not sufficient to cause a

sharp reduction in viral loads, particularly in the lungs, which is the organ with the highest viral load per gram of tissue.

Although Calpeptin is primarily given to rodents via subcutaneous or intraperitoneal administration, additional research is needed to establish its pharmacokinetics. This involves exploring diverse routes of administration and dosages according to the animal species. It is possible that a shorter treatment period with higher dosage, a more targeted application route such as nasal inhalation, or a combination of both could result in greater drug concentrations in the affected respiratory tissues and more positive outcomes.

Overall, SARS-CoV-2-infected hamsters treated with S-Calpeptin had lower viral loads in their trachea and lungs on day 5 post-infection compared to the untreated group. However, the decrease was only statistically significant in the trachea when measured with the TCID50 technique. Virus isolation in Vero cells only detects infective virions, whereas the RT-qPCR detects both free genomic RNA and genomic RNA in viable or nonviable viral particles. It is therefore conceivable that Calpeptin treatment was more effective in reducing infective virions, because RNA, even though damaged, must be first degraded before the RT-qPCR will show decrease in its values. The absence of statistically significant decrease in viral load in the S-Calpeptin-treated group may have been influenced by the S-Calpeptin dose, animal group size, and intrinsic host variability.

Under drug design aspects, a higher selectivity against CatL might be achieved through structural optimization of Calpeptin. According to our X-ray crystal structures, chemical modifications

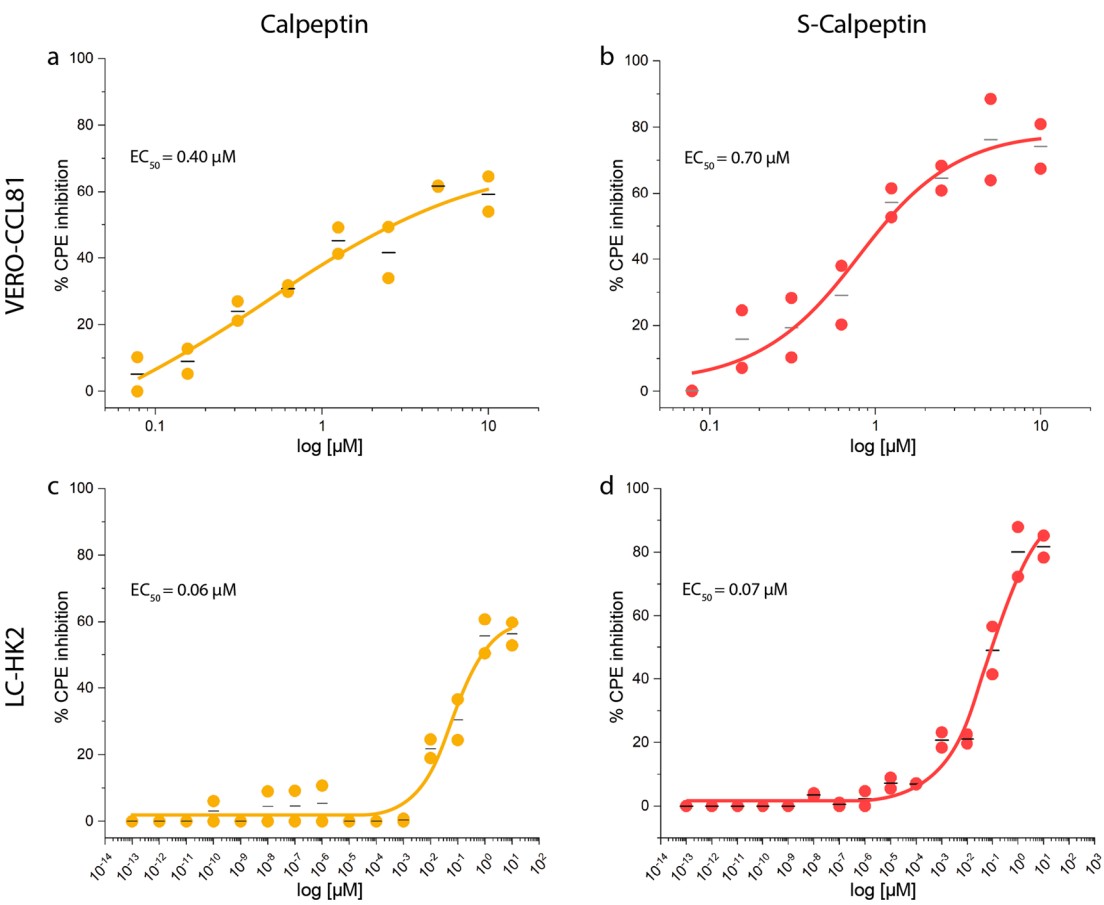

**Fig. 4 Calpeptin and S-Calpeptin inhibit SARS-CoV-2 mediated CPE.** Calpeptin or S-Calpeptin were added to VERO-CCL81 at 2-fold serial dilutions (**a**, **b**) and to human LC-HK2 at 10-fold serial dilutions (**c**, **d**). CPE inhibition was determined via CellTiter-Glo® Luminescent Assay. $EC_{50}$-values are shown. Individual data points (●) and the mean (——) of three replicates of two biological experiments for VERO-CCL81 cells (**a**, **b**) and LC-HK2 cells (**c**, **d**) are displayed. Compounds concentrations are presented in log scale for interpolation. $EC_{50}$ was calculated by fitting the data using GraphPad Prism version 8.0.

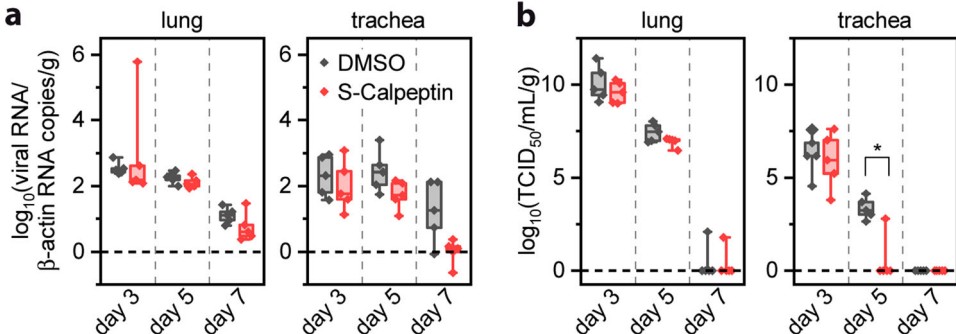

**Fig. 5 S-Calpeptin treatment reduces viral load in trachea in a hamster SARS-CoV-2 infection model.** Golden Syrian hamsters were intranasally inoculated with SARS-CoV-2 on day 0. Daily treatment started on day 1 with 1 mg/kg S-Calpeptin subcutaneously (s.c.) and 1:100 DMSO (dimethyl sulfoxide) as control. Viral load was measured at day 3, 5, and 7 post infection in lungs and trachea. **a** SARS-CoV-2 RNA level expressed as viral RNA copies/copies of β-actin/g of tissue. **b** Infectious viral particles expressed as 50% tissue culture infective dose/g of tissue. $n = 5$ animals per group. Values are expressed in median, interquartiles and range. Statistically significant differences ($p \leq 0.05$) were determined by two-sided Mann–Whitney $U$-test.

of Calpeptin to enable interactions with the S1' pocket of CatL and beyond may increase both overall affinity and isoform specificity[42]. However, cysteine cathepsins may play redundant roles in viral entry consistently with the inhibition of several cathepsins (Table 1). This suggests that strict selectivity of inhibitors for individual cathepsins is not the desired property in viral infections, but rather targeting of all the functionally redundant cathepsins.

Whereas the Delta variant of concern of the SARS-CoV-2 virus facilitates the cell entry mainly via the TMPRSS2-dependent cell surface pathway, the Omicron variant was shown to primarily enter the cell via the endosomal pathway[16,43]. The preference for the endosomal pathway including the proteolytic cleavage of S-protein by cysteine cathepsins of the Omicron variant is not surprising as it appears to be common to most human coronaviruses[16], possibly an outcome of convergent evolution. It

merits to mention that the Ebola virus also shares a similar activation pathway[44,45]. This indicates the potential of Calpeptin and other non-specific cysteine cathepsin inhibitors as possible broad-band antivirals. In addition, a combination therapy with cathepsin inhibitors and antivirals inhibiting the cell-surface entry mechanism, such as camostat blocking TMPRSS2[15], or a SARS-CoV-2 RNA polymerase inhibitor could further improve its effectiveness.

## Methods

**Synthesis of S-Calpeptin.** A mixture of Calpeptin (420 mg, 1.16 mmol, 1.0 eq.) and sodium bisulfite (121 mg, 1.16 mmol, 1.0 eq.) was dissolved in EtOAc/EtOH/H$_2$O (4:2:1, 15 mL) (Fig. S9a). The resulting reaction mixture was heated to 40 °C and stirred for 2 h. After cooling to room temperature, the solution was filtered, concentrated, and dried in vacuo to give the desired bisulfite adduct S-Calpeptin (524 mg, 1.12 mmol, 97% yield) as a white crystalline solid. The product was further characterized by UV (Fig. S9b), mass spectrometry (Fig. S9c, d), and nuclear magnetic resonance spectroscopy (Fig. S10).

**Protein purification.** Recombinant production of cathepsins was previously described for CatL and CatV[46], CatB[47], and CatK[48]. M$^{pro}$ was purified as previously described[49], with the only difference that 1 mM dithiothreitol in the storage buffer was replaced by 1 mM tris(2-carboxyethyl)phosphane-hydrochloride (TCEP).

**Cathepsin and M$^{pro}$ inhibition test.** Calpeptin, S-Calpeptin and GC-376 were tested for their inhibitory properties on human CatB, CatK, CatL, and CatV. All experiments were performed in solution of 50 mM sodium acetate, pH 5.5, 50 mM NaCl and 5 mM DTT. Measurements were taken at 37 °C in 96-well black flat bottom microplates (Greiner, Germany) using Tecan INFINITE M1000 pro plate reader (Tecan, Switzerland) with excitation and emission wavelengths of 370 and 460 nm, respectively[50].

For initial screening, cathepsins (1 or 10 nM) were mixed with different concentrations of inhibitors (50 μM–1 nM) to determine the range of their inhibition. For cathepsin—inhibitor pairs that exhibited inhibition in high nanomolar or micromolar range, $K_i$ was determined using 10 nM cathepsin solutions with several inhibitor and substrate concentrations. For those showing inhibition in pM range, cathepsins (0.33 nM) were incubated with 15 inhibitor concentrations of 1.8-fold dilution series (0.01–37.5 nM) and $K_i$ was calculated using Morrison equation.

To check whether the covalent bond between cathepsins and inhibitors is reversible, we incubated 1 nM CatL with S-Calpeptin and Calpeptin (concentration range 1–80 nM) for 60 min at 37 °C. We observed no increase in relative inhibition over time, thus concluding that the covalent bond between cathepsins and inhibitors is reversible.

Inhibition assays of M$^{pro}$ by Calpeptin, S-Calpeptin and GC-376 were performed in 20 mM HEPES, pH 7.3, 1 mM EDTA, 100 mM NaCl and 1 mM DTT at 37 °C. $K_i$ for S-Calpeptin and Calpeptin were determined using 100 nM M$^{pro}$ and varying inhibitor/prodrug concentrations (0.4–97 μM) and substrate concentrations. Inhibition by GC-376 was determined using 100 nM M$^{pro}$ and varying inhibitor concentrations (12.5–800 nM).

**Cell culture and virus production with SARS-CoV-2-GFP.** VERO E6 cells and their respective culturing conditions was described by Stukalov et al.[51]. All cell lines were tested to be mycoplasma-free. For SARS-CoV-2-GFP strain[52] production VERO E6 cells (in DMEM, 5% FCS, 100 μg/mL Streptomycin,

100 IU/mL Penicillin) were inoculated with virus stock at an MOI of 0.05. After an incubation period of 60 h (37 °C, 5% CO$_2$), virus-containing supernatant was collected, spun twice (1000 × g, 10 min) and stored at −80 °C. For determination of viral titers, a plaque assay was conducted. Confluent monolayers of VERO E6 cells were inoculated with fivefold serial dilutions of virus supernatants for 1 h at 37 °C. Hereafter, virus inoculum was discarded, serum-free MEM (Gibco, Life Technologies) containing 0.5% carboxymethylcellulose (Sigma-Aldrich) was added and incubated for 48 h (37 °C, 5% CO$_2$). After fixation with 4% formaldehyde (20 min at room temperature), cells were washed extensively with 1X PBS and stained with 1% crystal violet and 10% ethanol in H$_2$O for 20 min After another extensive washing step with 1× PBS, plaques were counted, and the virus titer was calculated.

**Cell lines and cell culture.** VERO-CCL81 cell line was obtained from ATCC (ATCC® CCL-81™). LC-HK2 cell line was derived from a tumor induced in nude rat by the inoculation of LC-HK1 cells which was spontaneously established from an explant of a cervical human non-small cell lung cancer metastasis and exhibits similarity to the original lung tumor characteristics, both morphologically and biochemically[53–55]. The LC-HK2 cells have pleiomorphic morphology, maintaining a small population of mono- or multinucleated giant cells and are aneuploid presenting hyperdiploid DNA content, as established for a human tumor lineage. In addition, LC-HK2 co-express both cytokeratin and vimentin intermediate filaments and are rich in actin microfilaments organized in stress fibers and network pattern exhibiting clusters[53–55].

This enables us to identify phenotypes, cellular and molecular changes that accompany the infection process, similar to those that are observed in vivo. Since LC-HK2 holds the biochemical and physiological properties of human lung tissue, it was used in our studies as a human cellular model for SARS-CoV-2 infection that exhibits the expression of genes of human lung tumoral cells and therefore is predicted to express both TMPRSS2 and CatL and support SARS-CoV-2 infection.

Both VERO-CCL81 and LC-HK2 cell lines were cultivated using Dulbecco's modified Eagle's medium (DMEM) supplemented with 10% fetal bovine serum (FBS), in an incubator at 37 °C and 5% CO$_2$ atmosphere.

**Analysis of TMPRSS2 expression by Western blot.** Cells LC-HK2 and VERO-CCL81 were washed three times with PBS, treated with trypsin to remove cell surface proteins and lysed with 1% Triton X-100 in PBS. Insoluble debris was removed by centrifugation, and the protein concentration was determined using BCA protein assay Pierce BCA Protein Assay Kit (Thermo Fisher Scientific), according to the manufacturer's instructions. Cell lysates were immediately diluted in sample buffer in the presence of reducing agent 5% (v/v) β-mercaptoethanol and incubated at 95 °C for 5 min to complete protein denaturation. The proteins were resolved by 10% SDS-PAGE and transferred to nitrocellulose membrane (Armesham GE Healthcare). TMPRSS2 was probed with primary antibody rabbit anti-TMPRRS2 (Sigma-Aldrich; ZRB1633, 1:1000 dilution) into PBS 0.05% tween 20 and 0.01% BSA, while α-tubulin was probed with anti-α-tubulin (Thermo Fisher Scientific; MA1-19401, 1:1000 dilution) into PBS 0.05% tween 20 and used as loading control. The membranes were incubated overnight at 4 °C and the binding of primary antibody was detected using horseradish peroxidase (HRP)-conjugated anti-rabbit (Cell Signaling; 7074S, 1:7500 dilution) or HRP-conjugated anti-mouse (KPL—SeraCare; 041806, 1:7500 dilution), respectively. Western blots were exposed to

chemiluminescent reagents WesternSure Premium Chemiluminescent substrate (926-95010, Li-cor Biosciences, Lincoln, USA) and analyzed by LI-COR Odyssey Imaging System, model 2802 (Li-cor Biosciences, Lincoln, USA).

**Cell viability assay**. VERO-CCL81 was seeded in 96-well plates at a density of $1.5\times10^4$ cells/well, while LC-HK2 was seeded at a density of $3\times10^4$ cells/well, following 24 h incubation at 37 °C and 5% $CO_2$ atmosphere. The cell culture media was replaced by serial dilutions of the compounds and the cell viability was determined 72 h post-treatment via CellTiter-Glo® Luminescent Cell Viability Assay (Promega), following the manufacturer's instructions. Luminescent signal was recorded using a CLARIOstar multi-mode microplate reader (BMG Labtech, Germany). Graphs were generated using GraphPad Prism software version 8.0 (La Jolla, CA, USA, www.graphpad.com). Samples deemed to be technical failures and extreme outlier were removed.

**Viral infection**. VERO-CCL81 was seeded in 96-well plates at a density of $1.5\times10^4$ cells/well, while LC-HK2 was seeded at a density of $3\times10^4$ cells/well following 24 hours incubation at 37 °C and 5% $CO_2$ atmosphere. The cells were pre-treated for 2 h with serial dilutions of compounds in fresh DMEM supplemented with 2.5% FBS. The compounds were removed, and SARS-CoV-2 strain[32] diluted in DMEM supplemented with 2.5% FBS, was added to the VERO-CCL81 cells at a MOI of 0.01 and to LC-HK2 cells at M.O.I of 0.05, allowing absorption for 1 h. The viral inoculum was removed, and cells were gently washed with phosphate-buffered saline (PBS) without calcium and magnesium. Fresh DMEM supplemented with 2.5% FBS containing serial dilutions of compounds was added back onto the cells. VERO-CCL81 was incubated for further 48 h, while LC-HK2 was incubated for further 72 h post-infection to assess viral loading or cytopathic effect. All SARS-CoV-2 infections were performed in a biosafety level 3 laboratory at the Institute of Biomedical Sciences, University of São Paulo, Brazil.

**Viral loading determination via RT-qPCR**. For viral loading evaluation, viral RNA was purified from cellular supernatant using MagMAX™ Viral/Pathogen Nucleic Acid Isolation Kit (Thermo Fisher Scientific) and the samples were processed using the semi-automated NucliSENS® easyMag® platform (bioMérieux, Lyon, France), following the manufacturer's instructions. The detection of viral RNA was carried out on a QuantStudio™ 3 Real-Time PCR System (Thermo Fisher Scientific) using the AgPath-ID™ One-Step RT-PCR Kit (Thermo Fisher Scientific) and a sequence of primers and probe for E gene[56]. The viral titers were calculated using a standard curve generated with serial dilutions of a template of known concentration and expressed in tissue culture infectious dose ($TCID_{50}$)/mL. Infected cells with 0.5% DMSO were used as control.

**Cytopathic effect inhibition assay**. When cytopathic effect occurs due to viral infection, ATP depletion can be measured and correlated with the viral burden[57]. The cytopathic effect following 48 h post-infection of VERO-CCL81 and 72 h post-infection of LC-HK2 was measured via CellTiter-Glo® (CTG) Luminescent Cell Viability Assay (Promega), following manufacturer's instructions. Luminescent signal was recorded using a CLARIOstar multi-mode microplate reader (BMG Labtech, Germany). Percent CPE inhibition was defined as [(test compound − virus control)/(cell control − virus control)] × 100[57].

**Fluorescence microscopy and image analysis**. LC-HK2 was cultivated on coverslips (13 mm diameter) to approximately 80%

confluence. The cells were pre-treated for 2 h with 1.5 μM of S-Calpeptin diluted in DMEM supplemented with 2.5% FBS. The compounds were removed, and SARS-CoV-2 strain diluted in DMEM supplemented with 2.5% FBS was added to LC-HK2 cells at M.O.I of 0.05, allowing absorption for 1 h. The viral inoculum was removed, and cells were gently washed with PBS without calcium and magnesium. The cells were incubated in the continued presence of 1.5 μM of S-Calpeptin diluted in DMEM supplemented with 2.5% FBS and were examined by fluorescence microscopy at the time points 32, 40, 48, 56, 64 and 72 h post-infection.

For this purpose, the cells were fixed and permeabilized with 3.7% formaldehyde solution (Sigma-Aldrich, F1635) containing 0.2% Triton X-100 in PBS 1×. The fixed specimens were washed with PBS-T (PBS 1×, 0.05% Tween 20) and blocked in PBS-AT (3% BSA, 0.5% Triton X-100 in PBS 1×) at room temperature for 30 min. The cells were incubated for 1 h at room temperature with primaries antibodies diluted in PBS-AT: rabbit anti-cathepsin L (Sigma-Aldrich—ZRB1636, 1:100 dilution) and mouse anti-SARS-CoV-2 spike (Thermo Fisher Scientific—MA536245, 1:200). Hoechst 3342 (Thermo Fisher Scientific—H1399, 10 μg/mL final concentration) was used to counterstain the DNA. The cells were gently washed with PBS-T and incubated with respective secondary antibodies diluted in PBS-AT: chicken anti-mouse AlexaFluor-488 (Thermo Fisher Scientific—A-21200, 1:500 dilution) and donkey anti-rabbit Alexa-Fluor-568 (Thermo Fisher Scientific—A10042, 1:500 dilution). The coverslips were mounted on glass slides using ProLong Gold mounting media (Thermo Fisher Scientific). From the coverslips of LC-HK2 a series of Z stack images were captured in 0.5 μm-thick sections using a ZEISS AxioObserver Z1 equipped with ApoTome2 and bi-gas incubation chamber with a ×100 oil-immersion objective. The images were captured using fluorescence range intensity adjusted identically within each experimental series. The entire fixed cell volume was displayed as 2D maximum projections using Image J FIJI (National Institutes of Health) or processed for 3D rendering using Zen 2.6 blue software (ZEISS).

Vesicles size quantifications were obtained from 2D maximum projections of Z stacks exported as grayscale TIFs (16-bit depth) using Automated Image Analysis in Zen 2.6 blue software (ZEISS). Compartments in diameter were generated from thresholding adjustments during overall segmentation. Any objects not meeting the threshold criteria in the segmentation were identified and manually excluded from analysis. To avoid artifact formation, only identified compartments with diameter range from 0.2 to 1 μm, which correspond to late endosomes or lysosomes[58], were accepted in the output.

**Infection assays for inhibitor screening**. VERO E6 were seeded at a density of 10,000 cells per well in 96-well plates. After an incubation time of 24 h, cells were inoculated with SARS-CoV-2-GFP (MOI 0.05) for 24 h. Using an EssenBioscience IncuCyte with IncuCyte 2020C Rev1 software, live-cell imaging was performed by taking pictures every 3 h (scan type: standard; image channels: Phase, Green to detect GFP, Red to detect RFP; objective: ×10). Integrated intensity of detected signal in the green channel was calculated by the IncuCyte 2020C Rev1 software.

**X-ray crystallography**. M$^{pro}$ was crystallized with compounds (Calpeptin, S-Calpeptin and GC-376) by adding 0.24 μL protein solution (6.25 mg/mL) in 20 mM HEPES buffer (pH 7.8) supplemented with 1 mM TCEP, 1 mM EDTA and 150 mM NaCl, with 0.21 μL reservoir solution containing 100 mM MIB buffer (malonate, imidazole, boric acid) pH 7.5, 25% PEG 1500 (*w/w*)

and 5% (*v/v*) DMSO, and 0.05 μL reservoir solution containing microseeds obtained by mechanical fragmentation with glass beads (Jena Bioscience). In the case of co-crystallization with Calpeptin, the DMSO contained 5 mM of the respective compound. GC-376 and S-Calpeptin was soaked into crystals, using 5 mM compound concentration and 1 h incubation. CatL was crystallized after removing the protection group S-methyl methanethiosulfonate from the active site Cys25 in the presence of 5 mM DTT. The glycosylation at the position Thr110 was changed by mutation to an alanine to avoid glycosylation. CatL at concentration of 7 mg/mL was equilibrated against 27% (*w/v*) PEG 8000, 1 mM TCEP and 0.1 M sodium acetate at pH 4.0 in MRC maxi plates by sitting drop vapor diffusion. Crystals grew to maximum size after approximately 3 days at 20 °C and were transferred to compound soaking solution, which contained 22% (*w/v*) PEG 8000, 1 mM TCEP and 0.1 M sodium acetate at pH 4.0 as well as 5% (*v/v*) DMSO and additional 10% (*v/v*) PEG 400 for cryoprotection. The final concentration of the compounds was adjusted to approximately 1 mM. After 12–16 h, the crystals were flash frozen in liquid nitrogen and measured at the PETRAIII synchrotron. The rod-like crystals were routinely measured at three positions and the datasets were merged to obtain complete datasets. All crystallographic data were processed with XDS and refinement was in general performed with phenix interspersed with manual model building in COOT. CatK with S-Calpeptin was crystallized in 30% of PEG-3350 and 0.2 M CaCl$_2$ at 293 K with sitting drop vapor diffusion experiments. CatV with Calpeptin was crystallized in 77% MPD and 23% of 60 mM TRIS, pH 8.0 at 278 K with sitting drop vapor diffusion experiments. The cathepsin structures were refined using MAIN[59]. Further structure analysis and visualization was done by using PyMOL and Chimera[60].

**In vivo studies, virus stock, and titration**. The SARS-CoV-2 (B.1.1.28, SARS-CoV-2/SP02/2020HIAE, GenBank MT126808.1) used in animal inoculations was first isolated in VERO E6 cells from a nasopharyngeal swab of one of the first patients reported with SARS-CoV-2 in Brazil[32]. The sample was confirmed to be free of 15 other viral agents (Endemic Coronavirus—CoV-NL63, -229E, -HKU1 and -OC43, Enterovirus, Influenza A and B, Parainfluenzavirus 1, 2, 3 and 4, Rhinovirus, Human Metapneumovirus, Respiratory Syncytial Virus, and Adenovirus) by RT-qPCR and SARS-CoV-2 presence was confirmed using a specific qRT-PCR assay[32,56].

For virus stock production, VERO CCL-81 cells were infected with a M.O.I. (multiplicity of infection) of 0.1 during 48 h, at 37 °C and 5% CO$_2$, and the virus was subsequently stored at −80 °C. For titration, virus samples were ten-fold serially diluted (10$^{-1}$–10$^{-12}$) in Dulbecco's Modified Eagle Medium (DMEM), with 2.5% of fetal bovine serum (FBS), and inoculated in sextuplicate on 96-well plates containing $2 \times 10^4$ VERO CCL-81 cells/well. After 72 h at 37 °C with 5% CO$_2$, the plates were microscopically inspected for CPE due to SARS-CoV-2 and the monolayer fixed and stained with Naphthol Blue Black (Sigma-Aldrich) solution (0.1% naphthol blue w/w with 5.4% of acetic acid and 0.7% of sodium acetate) for visual confirmation. Viral titer was calculated using the Spearman & Kärber algorithm[61] and expressed in TCID50 (Tissue Culture Infectious Dose)/mL. The isolate used in this study was on its third passage in VERO CCL-81 cells.

**S-Calpeptin toxicity experiment in Golden Syrian hamsters**. Twelve conventional male, Golden Syrian hamsters (*Mesocricetus auratus*; 6–8 weeks old) were acquired from the Instituto Gonçalo Muniz, Fiocruz, Salvador, Brazil and housed in the Animal

Facility of the Department of Preventive Veterinary Medicine and Animal Health of the College of Veterinary Medicine, University of São Paulo, Brazil. Animals were acclimatized for seven days, received food and water ad libitum and were divided into four groups (*n*=3 each), homogenized based on weight. Animals of each group received a daily subcutaneous dose of S-Calpeptin suspended in 1:100 dimethyl sulfoxide (DMSO) (1 mg/kg, 2 mg/kg, or 3 mg/kg of body weight) or a solution of 1:100 DMSO for seven consecutive days, starting at day zero, alternating the inoculation site from the right to the left flank in a maximum volume of 30 μL (Fig. S11a–f). Animals were checked and weighted daily. After seven days of inoculation, animals were euthanized with morphine (2.5 mg/kg) subcutaneously followed by an intraperitoneal overdose of ketamine (600 mg/kg) and xylazine (30 mg/kg). Immediately following cardiac arrest, total blood was collected via cardiac puncture and placed in microtubes containing EDTA (BD Biosciences, USA). Animals were necropsied and samples of the brain, liver, pancreas, gastrointestinal tract, kidney, lungs, spleen, and trachea were collected and fixed in 10% buffered formalin. Total blood was then subjected to a complete blood cell count at the private veterinary clinical pathology laboratory Lab&Vet, São Paulo, Brazil. All procedures were approved by the Committee on Animal Use and Experimentation from the College of Veterinary Medicine, University of São Paulo, Brazil (protocol # 8711260321).

**Anti-viral experiment with Golden Syrian hamsters**. A total of 36 conventional male, Golden Syrian hamsters (*Mesocricetus auratus*; 6–8 weeks old) were acquired from the Instituto Gonçalo Muniz, Fiocruz, Salvador, Brazil and housed at the Animal Biosafety Level 3 Laboratory of the Department of Parasitology, Institute of Biomedical Sciences, University of São Paulo, Brazil. Animals were acclimatized for seven days and kept individually in microisolators with food and water ad libitum. Animals were checked and weighted daily.

Hamsters were separated into four groups (G1-G4), homogenized based on weight. On day zero (infection day), all animals were initially anesthetized with 100 mg/kg of ketamine and 10 mg/kg of xylazine intraperitoneally. Hamsters from G1 (*n*=15) and G2 (*n*=15) were then inoculated intranasally with 10$^5$ TCID50 of SARS-CoV-2 (in 50 μL of DMEM, 2.5% FBS), while hamsters from G3 (*n*=3) and G4 (*n*=3) were inoculated intranasally with 50 μL of DMEM, 2.5% FBS. From day 1 to day 7 post infection (p.i.), animals from G1 and G3 received 1 mg/kg of S-Calpeptin diluted in 1:100 DMSO subcutaneously once a day (volume ranged from 16-26 μL/animal), while animals from G2 and G4 received 16-26 μL of 1:100 DMSO daily, also subcutaneously (Fig. S6a, b).

On days 3, 5, and 7 p.i., subgroups of 5 animals each from G1 and G2 were euthanized using 5 mg/kg of morphine subcutaneously followed by 600 mg/kg of ketamine and 30 mg/kg of xylazine intraperitoneally. Animals from G3 (n=3) and G4 (n=3) were euthanized with the same protocol on day 7 p.i. (Fig. S6a). Hamsters were necropsied and samples from trachea and lungs were collected for viral load determination (in DMEM, 2% FBS with 10.000 U/mL of penicillin, 10 mg/mL of streptomycin, 25 μg of amphotericin B/mL, and 2 mm glass beads) and histopathology (in 10% buffered formalin). Organ samples were individually weighted and rapidly frozen in liquid nitrogen and transferred to −80 °C for posterior analysis. Clean, sterile instruments were used between organ collection to avoid viral cross-contamination. All five lung lobes (right cranial lobe, right middle lobe, right caudal lobe, accessory lobe, and left lung) were equally represented in all samples and individually identified to be evaluated by histopathology analysis. All procedures were

approved by the Committee on Animal Use and Experimentation from the Institute of Biomedical Sciences, University of São Paulo, Brazil (protocol # 9498230321).

**Viral load quantification from tissue samples**. The viral load was quantified by determining the TCID50/g of tissue and the number of viral RNA copies/β-actin RNA copies/g of tissue[62]. Briefly, all samples of trachea and lungs were thawed and subjected to disruption using 2 mm glass beads in a TissueLyser II equipment (Qiagen, Germany) at 30 Hz for 2 min twice, followed by centrifugation at 13,000 rpm (Eppendorf 5804R centrifuge) for 1 min. The supernatants were used for viral load quantification by TCID50 determination and for viral RNA copies determination via quantitative reverse transcription PCR (RT-qPCR). For TCID50 determination, the samples were serially diluted ten-fold ($10^{-1}$–$10^{-12}$) with DMEM containing 2.5% FBS and inoculated in six replicates in 96-well plates containing VERO CCL-81 at a density of $5 \times 10^4$ cells/well. After incubation at 37 °C with 5% $CO_2$ for 72 h, the plates were microscopically inspected for CPE caused by SARS-CoV-2. The monolayers were fixed and stained with a Naphthol Blue Black (Sigma-Aldrich) solution (0.1% naphthol blue w/w with 5.4% acetic acid and 0.7% sodium acetate) and analyzed to confirm the results. The viral titer was calculated using the Spearman & Kärber algorithm and reported in TCID50/mL. For RT-qPCR, total nucleic acids were extracted using a Magmax Core Kit with a MagMAX Express Magnetic Particle Processor (Thermo Fisher Scientific) according to the manufacturer's instructions. The detection of SARS-CoV-2 RNA was performed based on a previously described protocol using a one-step RT–qPCR assay kit (AgPath-ID™ One-Step RT–PCR Reagents, Applied Biosystems Inc.) and an ABI 7500 SDS real-time PCR machine (Applied Biosystems). The duplex reaction was performed using specific primers for the E gene of SARS-CoV-2 and primers for β-actin gene (ActB Rv 5' CAC CAT CAC CAG AGT CCA TCA C 3', ActB F CTG AAC CCC AAA GCC AAC; ActB_P- HEX TGT CCC TGT ATG CCT CTG GTC GTA ZEN/IOWA BLACK) that was used as a housekeeping gene. The number of RNA copies/mL was quantified based on a standard curve obtained by serially diluting a synthetic dsDNA sequence (Gene Blocks, IDT) corresponding to the amplification fragment of the target gene. Data were obtained from three replicates in one biological experiment.

**Histopathology**. Organ samples from the toxicity and the antiviral experiments were collected on necropsy and fixed in 10% buffered formalin for 24 h. Three to five sections from each organ were obtained and samples were routinely processed for histological examination. Examination of hematoxylin and eosin-stained slides was done blindly by a board-certified anatomic pathologist on a Nikon® E200 optical microscope. For the antiviral experiment, lung lesions were semi-quantitatively evaluated based on parameters described in the literature[63–65] and according to an in-house developed scoring system (Table S1). A score was attributed to each parameter for each lung lobe of every animal. The final score of each lung parameter was constituted by the sum of scores attributed for each lung lobe. Therefore, the maximum value each lung parameter was able to reach was 15. For the trachea, a score of 0 was given to normal trachea, whereas scores of 1, 2 or 3 were given to mild, moderate, and severe tracheitis, respectively.

**Statistics and reproducibility**. In the enzyme inhibition assays, the initial velocities were calculated from the initial linear portions of their curves, assuming steady-state kinetics. IC50 and $Ki$ values were calculated using GraphPad Prism software. Z-FR-AMC (CatK and CatV), Z-RR-AMC (CatB and CatL), and QS1

or Acetyl-VKLQ-AMC ($M^{pro}$) substrates were used to monitor reactions. These measurements were done in four independent duplicates or triplicates, respectively.

The $EC_{50}$ of viral loading determination via RT-qPCR were calculated by fitting the data using GraphPad Prism version 8.00 (La Jolla, CA, USA, www.graphpad.com). Samples deemed to be technical failures and extreme outlier were removed.

For the cytopathic effect, the $EC_{50}$ values were fitted by sigmoidal function using GraphPad Prism version 8.00 (La Jolla, CA, USA, www.graphpad.com). Samples deemed to be technical failures and extreme outlier were removed.

In the Fluorescence microscopy evaluation, the $P$-values were generated by unpaired t-test using GraphPad Prism version 8.00 (La Jolla, CA, USA, www.graphpad.com). Samples deemed to be technical failures and extreme outlier were removed. $p < 0.05$ was considered as statistically significant.

At the Infection assay for inhibitor screening, the data were fitted to a four-parameter logistic function to derive EC50 values using software Origin 2021b.

Data from hamster weight loss during the infection period was subjected to a normality test according to D'Agostino & Pearson and then analyzed using one-way ANOVA followed by Tukey HSD. Viral load and histopathology scores were compared among groups using Mann–Whitney $U$-test or Kruskal–Wallis with Dunn test for pairwise comparisons. Results were considered statistically significant when $p \leq 0.05$ and analyses were performed in GraphPad Prism (version 9.1.1).

**Reporting summary**. Further information on research design is available in the Nature Portfolio Reporting Summary linked to this article.

## Data availability

The coordinates and structure factors of all described crystal structures are deposited in the PDB with accession codes 7QGW, 7QKA, 7QKB, 7QKC, 7Z3T, 7Z3U, 7Z58, 8C3D. Source data for all graphs and plots in the article can be found in the Supplementary Data file. All other materials are available from D.T., C.W., or A.M. upon request.

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

## Acknowledgements

We acknowledge DESY (Hamburg, Germany), a member of the Helmholtz Association HGF, for the provision of experimental facilities. Parts of this research were carried out at PETRA III beamline P11. This research was supported in part through the Maxwell computational resources operated at Deutsches Elektronen-Synchrotron DESY,

Hamburg, Germany. We acknowledge the assistance of Marcel Lach. We acknowledge financial support obtained from the Helmholtz society through the projects FISCOV, SFragX and the Helmholtz Association Impulse and Networking funds InternLabs-0011 "HIR3X". This work was further supported through BMBF funded projects "ConScience" (project 16GW0277) and the Röntgen-Angstrom cluster project "X-ray drug design platform" (13K22CHB). Additional funding was received through the Cluster of Excellence "Advanced Imaging of Matter" of the Deutsche Forschungsgemeinschaft (DFG)—EXC 2056—project ID 390715994, via BMBF via projects 05K19GU4 and 05K20GUB. The Drąg laboratory is supported by the "TEAM/2017-4/32" project, which is conducted within the TEAM program of the Foundation for Polish Science co-financed by the European Union under the European Regional Development Fund. The authors also thank the Fundação de Amparo à Pesquisa do Estado de São Paulo (FAPESP) (grants 2015/26722-8 (CW), 2020/12277-0 (EES), 2020/07251-2 (AMSG), 2020/09149-0, 2022/01812-8, and 2021/02736-0) and the collaborative network between the Universities São Paulo (USP) and Hamburg (UHH) via the UHH-USP-FAPESP Sprint Project 2019 (FAPESP 2019/00899-0). We also acknowledge the Butantan Institute (M.V.d.A. fellowship) and the Brazilian Ministry of Education, CAPES (88887.508739/2020-00, 000). D.T. group is supported by the Slovenian Research Agency (ARRS; research program P1-0048, Infrastructural program IO-0048).

## Author contributions

Designed research: P.Y.A.R., S.G., H.N.C., C.B., G.E., A.M.d.S.G., D.T., C.W., A.M. Wrote manuscript: P.Y.A.R., S.G., S.F., S.L.L., W.B., W.H., A.M.d.S.G., D.T., A.M. Sample preparation: P.Y.A.R., S.F., J. Lieske, J. Loboda, A.R.M., K.K., A.U., N.L., A.S., B.S., H.B., T.B. Conducted animal study: E.E.d.S., V.F.B., J.K., M.V.d.A., T.T.S.-P., A.F.d.S.F., M.S.T., E.L.D., P.R.G., M.B.H., A.M.d.S.G. Cell infection study: E.E.d.S., A.H., G.M.S., L.F.-A., D.B.L., G.E. X-ray data collection and analysis: P.Y.A.R., S.G., S.F., J. Lieske, W.E., J. Loboda, W.H., D.T. Performed and analyzed enzyme inhibition study: J. Loboda, A.H., K.K., A.U., N.L., A.S., D.T. Preparation and analysis of S-Calpeptin: M.H., R.C. Provided analysis resources: W.R., M.D.

## Funding

## Competing interests

The authors declare no competing interests.
