## [Peer Review File · Communications Biology]

Reviewers' comments:

Reviewer #1 (Remarks to the Author):

In this study, the authors confirmed that Calpeptin is a cysteine cathepsin inhibitor, its dual-targeting effect, and determined a series of crystal structures of Calpeptin, its sulfonated prodrug S-Calpeptin and the chemically similar GC-376 compound bound with Mpro, CatL, CatK, and CatV, and analyzed their interactional mode. They found that S-Calpeptin was a promising compound in cellular infection assay, and test its efficacy in the treatment of COVID-19 in a hamster model.

1. The inhibition of Mpro is in the micromolar range for Calpeptin and nanomolar range for GC-376. Do their aldehyde groups form covalent bond with Mpro?
2. Please test the binding affinity of Calpeptin, S-Calpeptin and GC-376 to CatK, CatL and Mpro.
3. Please explain the difference of Calpeptin in cathepsin and Mpro inhibition from their complex structures.
4. Don't overemphasize the advantage in targeting host proteins.

Reviewer #2 (Remarks to the Author):

The manuscript describes the dual inhibition effect of calpeptin, inhibiting both SARS- CoV-2 Mpro and host cathepsins. The authors have used X-ray crystallography to determine the complex structures of calpeptin, S-calpeptin, and GC-376 with cathepsin to understand the mechanism of action of these compounds. They have further tested the efficacy of these compounds in both cells based assays and infected Golden Syrian hamsters.

Comments:

Calpeptin is a calpain inhibitor, moreover, a structural analog of GC-376, which is known to inhibit Mpro of SARS-CoV-2 along with other calpain inhibitors (II, XII). Calpeptin inhibition of cathepsins and Mpro is already demonstrated by Jameel Inal et al. (clinical science, 2022), and Moin U. Fareed et. Al., (Am J Physiol Regul Integr Comp Physiol). A similar kind of study with GC-376 (Similar to calpeptin) and Calpain Inhibitor II, XI against both Mpro and Cathepsin L is already shown by Michael Dominic Sacco (Sacco et. al., Science) where they have reported the dual inhibition efficiency of these compounds.. The complex crystal structure of SARS-CoV-2 Mpro with GC-376 was also reported by Chunlong Ma (Chunlong Ma et al., Cell research).

Therefore, the present study does not have any significant advancement over the already published papers in this field to be considered for publication in Nature Communications Biology.

Other comments:

There are a lot of discrepancies/mislabelling between structure figures and their details in the results section. For example, Gly68/Gln19/Ala135 are mentioned in the text; however, they are not shown/labeled in the figure.

The authors should discuss their animal model results properly, explaining the possible reasons behind their observations. For example, why do they see discrepancies in the drug response between the trachea and the lung? Why, although viral RNA copies are not significantly changed, TCID50 change significantly after treatment with the inhibitor?

Fig S2H typos "activated"

Line 143...What do authors mean by tightly constrained binding pocket in Mpro leading to high sequence specificity? Mpro can also recognize peptides of different sequences. The authors should clarify this.

Reviewer #3

The manuscript entitled "Calpeptin is a potent cathepsin inhibitor and drug candidate for SARS-CoV-2 infections" by Reinke et al. reports on the results of a thorough *in vitro*, *in cellulo*, and *in vivo* study on Calpeptin as CatL inhibitor potentially useful to treat SARS-CoV-2 infections. I have very few considerations as the manuscript is indeed well crafted and significant experiments were attained to support the final conclusions. In particular, the authors convincingly show that Calpeptin is a proficient CatL inhibitor and that its inhibition does not lead to major acute effects in mice models while leading to an evident anti-SARS-CoV-2 activity.

As outlined by the authors, this MoA has advantages over targeting viral targets. On the other hand, Mpro inhibitors have entered the clinical practice and I believe that the concurrent targeting of hCatL and SARS-CoV-2 Mpro might result in a synergistic effect. Thus I would suggest the authors tried the combination between calpeptin and nirmatrelvir in infected cells and check if a synergistic effect is obtained.

Reviewer #4 (Remarks to the Author):

This manuscript described the antiviral activity and the mechanism of action of calpeptin and its prodrug S-calpeptin in inhibiting SARS-CoV-2. Calpeptin and S-calpeptin were shown to be dual inhibitors of SARS-CoV-2 main protease and the host cathepsins L, B, K, and V in the enzymatic assay. Cathepsins have been shown to be important host proteases in SARS-CoV-2 cell entry. Both compounds also inhibited SARS-CoV-2 replication in VERO-CCL81 and human LC-HK2 cells. X-ray crystal structures of calpeptin and S-calpeptin in complex with Mpro and cathepsin L have been solved, revealing similar binding poses and covalent conjugation with the catalytic cysteines. Furthermore, S-calpeptin also showed *in vivo* antiviral efficacy in SARS-CoV-2 infection hamster model and reduced viral load at day 5. Overall, the results are significant, especially the X-ray crystal structures, which provide a structural basis for the dual inhibitory mechanism. I strongly support the publication of this ms after proper revision. Comments are:

1. For the description of Mpro drug resistant mutants, the following key reference is missing: bioRxiv. 2022 Sep 6;2022.06.28.497978. doi: 10.1101/2022.06.28.497978.
2. "Such a dual-targeting approach has been reported for several calpain inhibitors^{3,18–20}"
Comment: the following key references of dual Mpro and cathepsins should be added:
J. Am. Chem. Soc. 2022, 144, 46, 21035–21045 (dual inhibitor with *in vivo* antiviral efficacy)
Sci. Adv. 2020, 6, eabe0751 (structural basis for the dual inhibition).
3. "In addition, the aldehyde warhead is sulfonated in GC-376, which is expected to increase the stability in the organism by protecting the aldehyde warhead from metabolism as demonstrated for Norovirus 3CLpro inhibitors²²"
Comment: Another advantage of bisulfide is the solubility.
4. Have the authors tested Calpeptin and S-calpeptin in TMPRSS2-expressing cell lines? It is expected that the antiviral potency will be drastically reduced.
5. Why the authors chose subcutaneously (s.c.) as the route of administration instead of oral or i.p.?
6. In Fig. S2, in the complex structure with Mpro, the Cbz group in S-calpeptin can adopt two different orientations. This has been observed for other Mpro inhibitors as well. See reference below:
Sci. Adv. 2020, 6, eabe0751

We thank all four reviewers for their efforts in reviewing our manuscript. All comments were very helpful. In the revised manuscript we have addressed these comments and performed additional experiments to support our findings. We believe that the manuscript has improved significantly through this review process.

Please find our detailed reply (blue text) to the reviewers' comments (highlighted in bold letters) below.

Reviewer #1 (Remarks to the Author):

In this study, the authors confirmed that Calpeptin is a cysteine cathepsin inhibitor, its dual-targeting effect, and determined a series of crystal structures of Calpeptin, its sulfonated prodrug S-Calpeptin and the chemically similar GC-376 compound bound with Mpro, CatL, CatK, and CatV, and analyzed their interactional mode. They found that S-Calpeptin was a promising compound in cellular infection assay and test its efficacy in the treatment of COVID-19 in a hamster model.

1. The inhibition of Mpro is in the micromolar range for Calpeptin and nanomolar range for GC-376. Do their aldehyde groups form covalent bond with Mpro?

In our and other published crystal structures of Calpeptin/GC-376 and derivatives bound to M^{pro} and CatL covalent bonds are observed between the protein's sulfur and the compound's oxygen with interatomic distances around 1.8 Å, which are unambiguously resolved by the electron density maps. We added the distance and the references in the Supplementary Methods text: l. 918f.: "Calpeptin binds covalently to Cys¹⁴⁵ of M^{pro} (PDB: 7AKU)³ with its aldehyde group, forming a thiohemiacetal with an interatomic S-O distance of 1.8 Å (Figure S2F)^{60,61}."

2. Please test the binding affinity of Calpeptin, S-Calpeptin and GC-376 to CatK, CatL and Mpro.

The usual biochemical measure for quantifying activity of reversible inhibitors is K_i , which is related to $\Delta G = -RT \ln K_i$. The K_i values from our inhibition experiments are shown in table 1. We also performed the inhibition assay with GC-376 and M^{pro}, and due to the sensitivity of the assay, we were only able to detect inhibition with an accuracy of <100 nM. This value has been added to table 1. In addition, we kept the corresponding literature value, which is consistent with our experiments.

3. Please explain the difference of Calpeptin in cathepsin and Mpro inhibition from their complex structures.

We address the differences of the protein-inhibitor complexes in the results section (X-ray structure analysis, l. 142ff.) and the supplemental section "Detailed description of the X-ray crystal structures of protease-inhibitor complexes" (l. 916ff.).

4. Don't overemphasize the advantage in targeting host proteins.

We agree with the reviewer that the targeting of host proteins is difficult and should be used under caution. Thus, we think short-termed inhibition of host proteins, which are essential for viral invasion, but not essential for short-term survivability of the host cell, is viable and valuable.

We have weakened the statement in the abstract (l. 72f.: “Despite a higher risk of side effects, an intrinsic advantage in targeting host proteins is their mutational stability in contrast to highly mutable viral targets.”) and further added the following sentences to the introduction: l. 87ff. “The inhibition of essential host proteins generally bears a risk of side effects. However, an acute life-threatening viral infection is different from a long-term chronic disease such as osteoporosis. Nevertheless, advantages and disadvantages of such an approach must be carefully considered.”

Reviewer #2 (Remarks to the Author):

The manuscript describes the dual inhibition effect of calpeptin, inhibiting both SARS- CoV-2 Mpro and host cathepsins. The authors have used X-ray crystallography to determine the complex structures of calpeptin, S-calpeptin, and GC-376 with cathepsin to understand the mechanism of action of these compounds. They have further tested the efficacy of these compounds in both cells based assays and infected Golden Syrian hamsters.

Comments:

Calpeptin is a calpain inhibitor, moreover, a structural analog of GC-376, which is known to inhibit Mpro of SARS-CoV-2 along with other calpain inhibitors (II, XII). Calpeptin inhibition of cathepsins and Mpro is already demonstrated by Jameel Inal et al. (clinical science, 2022), and Moin U. Fareed et. Al., (Am J Physiol Regul Integr Comp Physiol). A similar kind of study with GC-376 (Similar to calpepetin) and Calpain Inhibitor II, XI against both Mpro and Cathepsin L is already shown by Michael Dominic Sacco (Sacco et. al., Science) where they have reported the dual inhibiton efficiency of these compounds. The complex crystal structure of SARS-CoV-2 Mpro with GC-376 was also reported by Chunlong Ma (Chunlong Ma et al., Cell research).

Therefore, the present study does not have any significant advancement over the already published papers in this field to be considered for publication in Nature Communications Biology.

We agree with the reviewer that calpeptin was discovered as a calpain inhibitor, however we must disagree with the reviewer on the rest. In the manuscript we present novel X-ray structures of GC-376 bound to CatL and of Calpeptin bound to CatL, CatV, and CatK which provide evidence for cathepsin inhibition on atomic level for the first time. We further present the results of comparative enzymatic inhibition assays, allowing the direct comparison of these inhibitors, which was not possible with the literature values available so far. Although our hamster experiments with S-Calpeptin only show a significant effect on viral loads in the trachea, it generally demonstrates the potential of targeting cathepsins for the treatment of SARS-CoV-2 and possibly other viral infections that use the endosomal entry pathway. We therefore consider our results as a very valuable contribution to the field. The reference [Inal et al. 2022] mentioned by the reviewer was already cited, we have further added references [Fareed et al. 2006, l. 84; Sacco et al. 2020, l. 82; Ma et al. 2020, l. 82] as suggested by the reviewer to the manuscript where appropriate.

Other comments:

There are a lot of discrepancies/mislabelling between structure figures and their details in the results section. For example, Gly68/Gln19/Ala135 are mentioned in the text; however, they are not shown/labeled in the figure.

We thank the reviewer for pointing this out. We have carefully checked and corrected all figures for discrepancies and errors. The missing labels have been added to figure 2: l. 845.

We corrected the panel nomenclature in Figure S3: l. 1022ff.

The authors should discuss their animal model results properly, explaining the possible reasons behind their observations. For example, why do they discrepancies in the drug response between the trachea and the lung?

We appreciate the reviewer's concerns regarding our hamster experiments. In the supplementary text, we have presented the pre-experiment, which helped us determine the appropriate dose of S-Calpeptin for hamsters. We could not find any previous studies in the literature that described the administration of S-Calpeptin in hamsters; the studies were only conducted in mice. Hence, we were unaware of potential acute toxicities of S-Calpeptin in this species of rodent. Consequently, we tested three different doses of S-Calpeptin (1 mg/kg, 2 mg/kg, and 3 mg/kg) subcutaneously to hamsters for six days in a pre-experiment. Although the animals displayed no signs of acute toxicity, a complete blood cell count on day 6 showed that the treatment with 2 mg/kg and 3 mg/kg of S-Calpeptin led to marked leukopenia caused by lymphopenia and neutropenia, which was not seen in animals that received only 1 mg/kg of S-Calpeptin. For this reason, we restricted the administration of S-Calpeptin to 1 mg/kg in the viral challenge experiment.

We suspect that the modest decrease observed in the viral loads of hamsters treated with S-Calpeptin and the inconsistency between the lungs and trachea may be attributed to the dosage limitation enforced to prevent side effects. Although the viral load was lower in the treated group compared to the untreated group, it is possible that the distribution and levels of S-Calpeptin in the tissue were not sufficient to produce a sharp reduction in viral loads that is statistically significant, particularly in the lungs, which is the organ with the highest viral load per gram of tissue. We concur with the reviewer's suggestion that this needs to be more comprehensively explained in the text. Thus, we have incorporated the following details into the discussion section:

l. 245ff.: "Due to its broad-band inhibition of proteases, calpeptin is expected to cause several side effects during daily treatment. The dosage chosen for our hamster studies was a daily dose of 1 mg/kg s.c. over seven days with no apparent side effects, while higher doses (2 mg/kg or 3 mg/kg) resulted in blood leukopenia at the end of the seven-days protocol. The modest reductions in viral loads observed *in vivo* could be attributed to this limitation in dosage to avoid side effects. It is possible that the distribution and levels of S-Calpeptin in the tissue were not sufficient to cause a sharp reduction in viral loads, particularly in the lungs, which is the organ with the highest viral load per gram of tissue."

Why, although viral RNA copies are not significantly changed, TCID50 change significantly after treatment with the inhibitor?

To address this question, we must emphasize that the techniques employed differ in their ability to detect defective viral particles. The virus isolation by TCID50 detects infective virions only, i.e. virus particles that are able to infect cells. Conversely, the RT-qPCR detects free genomic RNA as well as genomic RNA incorporated in either viable or nonviable viral particles. Therefore, it is possible that the calpeptin treatment was more effective in reducing infective virions, whereas qPCR will not show a decrease in overall viral RNA levels. We have incorporated this information, along with other possible explanations, in the discussion section:

I. 259ff.: “Overall, SARS-CoV-2-infected hamsters treated with S-Calpeptin had lower viral loads in their trachea and lungs on day 5 post-infection compared to the untreated group. However, the decrease was only statistically significant in the trachea when measured with the TCID50 technique. Virus isolation in Vero cells only detects infective virions, whereas the RT-qPCR detects both free genomic RNA and genomic RNA in viable or nonviable viral particles. It is therefore conceivable that calpeptin treatment was more effective in reducing infective virions, because RNA, although not packed into viable virions, must be first degraded before the RT-qPCR will show decrease in its values. The absence of statistically significant decrease in viral load in the S- Calpeptin treated group may have been influenced by the S-Calpeptin dose, animal group size, and intrinsic host variability.”

Fig S2H typos “activated”

Figure changed accordingly.

Line 143...What do authors mean by tightly constrained binding pocket in M^{pro} leading to high sequence specificity? M^{pro} can also recognize peptides of different sequences. The authors should clarify this.

Here we compare the M^{pro} and CatL active sites. While the CatL active site is an open groove offering peptide backbone interactions, the M^{pro} active site is a closed pocket with constrained backbone interactions in addition to several strongly favored side chain pockets. M^{pro} is a chymotrypsin-like protease unrelated sequentially and structurally to cysteine cathepsins. As a consequence, also their active sites are different. Cysteine cathepsins have only one real substrate binding subsite in the shape of a pocket, namely S2, whereas the S1 subsite is pointing to solvent. In contrast, M^{pro} has a clear S1 pocket, as most of the other proteases have. The protein surface representation in figure S2 shows the comparison of CatL and M^{pro} substrate binding surfaces in detail. For further explanation of the active site differences among cysteine proteases the referee can reach out to our recent review (Tušar et al. *Int. J. Mol. Sci.* 2021, 22, 997. <https://doi.org/10.3390/ijms22030997>).

Reviewer #3 (Remarks to the Author):

The manuscript entitled “Calpeptin is a potent cathepsin inhibitor and drug candidate for SARS-CoV-2 infections” by Reinke et al. reports on the results of a thorough in vitro, in cellulo, and in vivo study on Calpeptin as CatL inhibitor potentially useful to treat SARS-CoV-2 infections. I

have very few considerations as the manuscript is indeed well grafted and significant experiments were attained to support the final conclusions. In particular, the authors convincingly show that Calpeptin is a proficient CatL and that its inhibition does not lead to major acute effects in mice models while leading to an evident anti-SARS-CoV-2 activity. As outlined by the authors, this MoA has advantages over targeting viral targets. On the other hand, Mpro inhibitors have entered the clinical practice and I believe that the concurrent targeting of hCatL and SARS-CoV-2 Mpro might result in a synergistic effect.

Thus, I would suggest the authors tried the combination between calpeptin and nirmatrelvir in infected cells and check if a synergistic effect is obtained.

Since a key feature of viral drug combinations is the potential for multiplicative or synergistic effects by targeting different targets and steps of the viral lifecycle, this is indeed a valuable suggestion for further research in the future. However, a comparably comprehensive analysis of the synergistic effect achieved by calpeptin in combination with nirmatrelvir extends beyond the purpose of this work.

Reviewer #4 (Remarks to the Author):

This manuscript described the antiviral activity and the mechanism of action of calpeptin and its prodrug S-calpeptin in inhibiting SARS-CoV-2. Calpeptin and S-calpeptin were shown to be dual inhibitors of SARS-CoV-2 main protease and the host cathepsins L, B, K, and V in the enzymatic assay. Cathepsins have been shown to be important host proteases in SARS-CoV-2 cell entry. Both compounds also inhibited SARS-CoV-2 replication in VERO-CCL81 and human LC-HK2 cells. X-ray crystal structures of calpeptin and S-calpeptin in complex with Mpro and cathepsin L have been solved, revealing similar binding poses and covalent conjugation with the catalytic cysteines. Furthermore, S-calpeptin also showed in vivo antiviral efficacy in SARS-CoV-2 infection hamster model and reduced viral load at day 5. Overall, the results are significant, especially the X-ray crystal structures, which provide a structural basis for the dual inhibitory mechanism. I strongly support the publication of this ms after proper revision. Comments are:

1. For the description of Mpro drug resistant mutants, the following key reference is missing:

bioRxiv. 2022 Sep 6;2022.06.28.497978. doi: 10.1101/2022.06.28.497978

We thank the reviewer for pointing out this and the following (see comment 2) additional references. This reference has been added (l. 79).

2. "Such a dual-targeting approach has been reported for several calpain inhibitors3,18–20"

**Comment: the following key references of dual Mpro and cathepsins should be added:
J. Am. Chem. Soc. 2022, 144, 46, 21035–21045 (dual inhibitor with in vivo antiviral efficacy)**

Sci. Adv. 2020, 6, eabe0751 (structural basis for the dual inhibition).

Both references have been added (l. 107).

3. "In addition, the aldehyde warhead is sulfonated in GC-376, which is expected to increase the stability in the organism by protecting the aldehyde warhead from

metabolism as demonstrated for Norovirus 3CLpro inhibitors²² "

Comment: Another advantage of bisulfide is the solubility.

We thank the reviewer for this remark. We added this information in the results section I.122ff: "In addition, the aldehyde warhead is sulfonated in GC-376, which is expected to increase the stability in the organism by protecting the aldehyde warhead from metabolism as demonstrated for Norovirus 3CLpro inhibitors²⁶ and, as additional advantage, to increase its solubility."

4. Have the authors tested Calpeptin and S-calpeptin in TMPRSS2-expressing cell lines? It is expected that the antiviral potency will be drastically reduced.

We tested inhibition by Calpeptin and S-Calpeptin in a parallel study in VERO-CCL81 cells, an established cell line for SARS-CoV-2 infection, and LC-HK2, a cell line that is predicted to express TMPRSS2 as it is derived from human lung tumoral and is therefore susceptible to SARS-CoV-2 infections, via the TMPRSS2 mediated surface entry pathway. To support this hypothesis, we performed a Western blot analysis, which confirmed that VERO-CCL81 cells barely express TMPRSS2 while LC-HK2 strongly express it, enabling the utilization of both surface and endosomal pathways to viral entry in these cells (new Figure S4).

Indeed, we observed similar inhibition for both cell lines with S-Calpeptin independent of presence of TMPRSS2. To better describe this observation the following section:

Original text, l. 179ff.: "Both were tested for inhibition of SARS-CoV-2 infections *in vitro*. In contrast to VERO E6 cells that do not express TMPRSS2, its parental cell line VERO-81 expresses TMPRSS2²⁶, enabling both the surface and endosomal entry pathway of the virus. Besides these two established cell lines for SARS-CoV-2 infection, we have used human non-small cell lung carcinoma LC-HK2 to mimic the cells that SARS-CoV-2 would first encounter when introduced into the lung."

was changed to

"Both were tested for inhibition of SARS-CoV-2 infections at the cellular level. We have used two cell lines for SARS-CoV-2 infection VERO E6 and the closely related VERO-CCL81 cells³⁰. While the former does not express TMPRSS2, the latter expresses TMPRSS2 at a very low level¹⁷ (Figure S4). Besides these two established cell lines for SARS-CoV-2 infection we have used human non-small cell lung carcinoma LC-HK2 to mimic the cells that SARS-CoV-2 would first encounter when introduced into the lung and enable both the surface and endosomal entry pathway of the virus. For this cell line we were able to detect strong TMPRSS2 expression in comparison with VERO-CCL81 (Figure S4)."

We added a new Figure S4 describing the Western blot experiment, showing the expression of TMPRSS2 in Vero-CCL81 and LC-HK2 cells. (l. 1031). The related experimental procedure was added to the supplemental Material and Methods section (l. 957 ff.)

Further insertion at l. 356: [... it was used in our studies as a human cellular model for SARS-CoV-2 infection] "that exhibits the expression of genes of human lung tumoral cells and therefore is predicted to express both TMPRSS2 and CatL and support SARS-CoV-2 infection."

5. Why the authors chose subcutaneously (s.c.) as the route of administration instead of oral or i.p.?

The subcutaneous (SC) route was selected for many reasons. First, along with the intraperitoneal (IP) route, it has been effectively employed in previous mouse studies investigating S-Calpeptin for other purposes. Second, it is a less stressful and lower-risk route for daily application in hamsters. These animals have a bulky gastrointestinal tract, the risk of perforation and peritoneal infection is high when administering drugs via IP route. Third, a daily intranasal instillation (not nebulization) requires anesthesia. In our experience, daily anesthesia is impractical for these animals and results in weight loss. Since weight is a key variable during experimental infection with SARS-CoV-2, any factor that affects weight gain has a negative impact on the study design. Fifth, as a proof of principle study, we initially selected routes commonly used in laboratory animals for S-Calpeptin rather than attempting to nebulize it. While we agree nebulization is a promising route of administration in this case, especially because of the side effects of systemic administration, further investigations are required to evaluate different routes and dosages of S-Calpeptin considering the animal model utilized. To address some of these concerns in the text, we also added the following sentences in the discussion section:

I. 253ff. “Although Calpeptin is primarily given to rodents via subcutaneous or intraperitoneal administration, additional research is needed to establish its pharmacokinetics. This involves exploring diverse routes of administration and dosages according to the animal species. It is possible that a shorter treatment period with higher dosage, a more targeted application route such as nasal inhalation, or a combination of both could result in greater drug concentrations in the affected respiratory tissues and more positive outcomes”.

6. In Fig. S2, in the complex structure with Mpro, the Cbz group in S-calpeptin can adopt two different orientations. This has been observed for other Mpro inhibitors as well. See reference below:

Sci. Adv. 2020, 6, eabe0751

We thank the reviewer to this remark. We added the reference and the following words “[...], a similar flexibility as observed for GC-376 analogs¹⁰.” (l. 937 ff.).

Further changes to the manuscript:

1. Due to a change in name of department, the affiliation 8 changed to ⁸Department of Cell and Developmental Biology, Institute of Biomedical Sciences, University of São Paulo, São Paulo, Brazil
2. The numbering of the figures was changed due to the addition of Figure S4.
3. “ebola” changed to “Ebola” at line 281
4. Funding chapter was modified at line 811 (“2022/01812-8” was added) and line 816 (“group” was added)
5. The word “animals” was added in line 883.
6. The word “more closely” was added in line 183.

REVIEWERS' COMMENTS:

Reviewer #1 (Remarks to the Author):

The author has already answered my question well. I think the new manuscript is ready for publication.

Reviewer #3 (Remarks to the Author):

The authors decided not to address the issue raised by the present Reviewer (testing the combination therapy with approved Mpro inhibitor). I do believe that this might have strengthened the findings of the paper. Nevertheless, it remains solidly experimentally substantiated that the sole Calpeptin treatment might have an antiviral effect. In this respect, I would endorse the manuscript publication on the Communications Biology journal.

Reviewer #4 (Remarks to the Author):

Comments from the previous review were properly addressed.